# Dietary Phosphorus as a Marker of Mineral Metabolism and Progression of Diabetic Kidney Disease

**DOI:** 10.3390/nu13030789

**Published:** 2021-02-27

**Authors:** Agata Winiarska, Iwona Filipska, Monika Knysak, Tomasz Stompór

**Affiliations:** Department of Nephrology, Hypertension and Internal Medicine, University of Warmia and Mazury in Olsztyn, 10561 Olsztyn, Poland; agatawiniarska@hotmail.com (A.W.); filipskaiwona@gmail.com (I.F.); makb@poczta.fm (M.K.)

**Keywords:** phosphorus intake, phosphaturia, hyperphosphatemia, diabetic kidney disease, chronic kidney disease, CKD progression

## Abstract

Phosphorus is an essential nutrient that is critically important in the control of cell and tissue function and body homeostasis. Phosphorus excess may result in severe adverse medical consequences. The most apparent is an impact on cardiovascular (CV) disease, mainly through the ability of phosphate to change the phenotype of vascular smooth muscle cells and its contribution to pathologic vascular, valvular and other soft tissue calcification. Chronic kidney disease (CKD) is the most prevalent chronic disease manifesting with the persistent derangement of phosphate homeostasis. Diabetes and resulting diabetic kidney disease (DKD) remain the leading causes of CKD and end-stage kidney disease (ESRD) worldwide. Mineral and bone disorders of CKD (CKD-MBD), profound derangement of mineral metabolism, develop in the course of the disease and adversely impact on bone health and the CV system. In this review we aimed to discuss the data concerning CKD-MBD in patients with diabetes and to analyze the possible link between hyperphosphatemia, certain biomarkers of CKD-MBD and high dietary phosphate intake on prognosis in patients with diabetes and DKD. We also attempted to clarify if hyperphosphatemia and high phosphorus intake may impact the onset and progression of DKD. Careful analysis of the available literature brings us to the conclusion that, as for today, no clear recommendations based on the firm clinical data can be provided in terms of phosphorus intake aiming to prevent the incidence or progression of diabetic kidney disease.

## 1. Phosphate Homeostasis

Phosphorus accounts for ~1% of whole body mass and is essential for cellular function. It is a main constituent of teeth and bones, but also critically contributes to the generation and storage of cell energy at the molecular level (in the form of ATP (Adenosine triphosphate)) [1]. Most body phosphorus is deposited in the bone as a hydroxyapatite. The gastrointestinal (GI) tract, bones and the kidneys are the key players in regulation of phosphorus homeostasis, whereas parathyroid hormone (PTH), fibroblast growth factor 23 (FGF23) and other phosphatonins, klotho protein and vitamin D remain key hormones precisely controlling this homeostasis. Dietary phosphate intake (especially with high-phosphate containing products such as processed food) is a critical factor determining its gut absorption, kidney reabsorption, bone deposition and tissue content. Since several toxicities can be attributed to the phosphate excess, fine-tuned regulation of phosphate balance is essential to keep normal body homeostasis. The amount of phosphorus absorbed from the GI, reabsorbed into the kidney and eliminated in the urine, as well as deposited in the bone tissue, must be precisely balanced in order to avoid potential toxicities. Phosphorus is absorbed from the GI lumen using sodium- dependent and sodium-independent pathways and is largely regulated by active forms of vitamin D [2,3]. Sodium-dependent absorption is provided mostly using NaPi-IIb (Npt2b) protein upon the control of calcitriol. Mechanisms of transcellular and paracellular, sodium-independent phosphate transport have not been fully elucidated. It seems that these pathways are calcitriol independent (which could explain increased absorption of phosphorus in patients with advanced chronic kidney disease, i.e., the condition of severe vitamin D depletion) [3,4]. Renal phosphate reabsorption is provided by sodium-dependent co-transporters localized in the luminal membranes of proximal tubular cells. They include: sodium-phosphate co-transporter type IIa (NaPi-IIa, Npt2a), IIc (NaPi-IIc, Npt2c) and sodium-potassium co-transporter type III (Pit-2, Ram-1) [5]. FGF (fibroblast growth factor) 23 and PTH (parathyroid hormone), key phosphaturic hormones increase phosphaturia by inhibition of Npt2a. FGF23, phosphatonin released by osteocytes, needs klotho protein as a co-factor to act on the proximal tubule phosphate absorption. Several other factors are involved in phosphate homeostasis in the kidney and include: growth hormone, insulin-like growth factor (IGF-1), insulin, thyroid hormones, secreted frizzled-related protein 4 (sFRp-4) and FGF 7. Phosphate (and calcium) deposition and resorption in and from the bones is also precisely regulated by several hormones acting on the osteoclasts and osteoblasts [3,6].

## 2. Abnormalities of Phosphate Homeostasis in Chronic Kidney Disease (CKD)

Mineral and bone disorders of chronic kidney disease (CKD-MBD) are considered the key homeostatic abnormalities of CKD. It should be emphasized that the dysregulation of calcium and phosphate balance develops far before the development of overt uremic toxicity (uremia)—it may be detected as early as in CKD stage 2 and then progress further with decreasing glomerular filtration rate (GFR). Most of the abnormalities (both at the laboratory and the clinical level) can already be identified in CKD stage 3b (i.e., when GFR falls to the range of 30–45 mL/min/1.73 m^2^). CKD-MBD affects almost all aspects of metabolism, beyond the bone turnover (historically, clinicians and researchers were focused mainly on this aspect of described abnormalities, referred to as “renal osteodystrophy”). Due to their abundance, advancement and clinical importance, cardiovascular (CV) consequences of CKD-MBD are now included in the definition of this clinical entity [7]. Phosphate retention directly translates into increased risk of CV events and mortality, as well as the risk of bone fractures and faster progression of CKD. Since the normal serum phosphate concentration is a homeostatic priority in order to avoid the consequences of hyperphosphatemia, it is kept within the normal range until the late stages of CKD owing to significantly diminished reabsorption in remaining (functioning) nephrons (i.e., markedly increased single nephron phosphaturia). It is achieved at the expense of significantly elevated serum FGF23 and PTH [8]. Since healthy kidney tissue is the most important source of klotho (co-factor of FGF23 receptor), CKD progression leads to the renal FGF-23 resistance, which additionally boosters synthesis and release of this phosphatonin. FGF23 at supraphysiologic concentrations (needed to maintain phosphaturia), interacting with FGF23 receptor without klotho, influences the tissues normally out of its control (for example triggers heart injury and hypertrophy and activates renin-angiotensin-aldosterone axis) [9]. Along with CKD progression (and progressive loss of nephrons), an excess of phosphaturic hormones fails to control phosphate level and hyperphosphatemia develops. Since phosphate intake is a potentially modifiable factor that impacts on phosphate homeostasis, the reduction of an excess phosphate intake with phosphate binders has long been recognized as one of the key therapeutic strategies in CKD. Indeed, many observational studies demonstrated the relationship between dietary phosphate and CV events and outcome [10]. On the other hand, it should be kept in mind that hypophosphatemia and low phosphate intake may also result in an increased mortality. Malnutrition seems the most obvious way to interpret such a finding. However, as shown by Chang et al., a relationship between low phosphate and mortality persists also after adjustment for nutritional markers, suggesting other possible mechanisms. Interestingly, the mortality increment curve was steeper for decreasing than for increasing phosphate level [11]. In this review we focused on the adverse effects of hyperphosphatemia and high-phosphate diet, but the opposite situation should not be lost from the reader’s perspective as an equal threat for the patients.

Recent trials however demonstrated that the correlation between phosphate intake and serum phosphate is relatively weak [12,13]. Selamet et al. analyzed the outcome of 795 patients with CKD stage G3a-G5 who were included into the Modification of Diet in Renal Disease (MDRD) trial (with mean follow-up of 16 years). The authors analyzed the relationship between phosphatemia, phosphate intake based on 3-day dietary recall and phosphate loss with 24-h urine and found that the correlation between these parameters is rather weak. Twenty four hour urine phosphate loss did not correlate with the risk of end-stage renal disease (ESRD), CV death or all cause-death, whereas a strong correlation was found between the serum phosphate and all-cause mortality [14]. This study confirmed the importance of hyperphosphatemia as predictor of CV and renal end-points but did not show the importance of phosphate intake as a contributor to hyperphosphatemia. The concept of preventing or treating hyperphosphatemia with dietary restrictions has also been challenged recently by the data suggesting that decreased phosphorus intake may significantly increase the efficacy of its intestinal absorption—the percentage or fraction of phosphate absorbed vs ingested (mechanism physiologically designed to keep normal bone homeostasis in periods of dietary phosphorus depletion). Such an increase in absorption may potentially counterbalance the potentially harmful effects of dietary phosphate deficiency on bone quality. These discoveries may in the future shift the main strategy of prevention and treatment of hyperphosphatemia from reduced intake and use of phosphate binders into the use of compounds that inhibit passive paracellular and active sodium-dependent transcellular phosphate transport in the intestinal epithelium. Tenapanor, the inhibitor of sodium—protein exchanger 3 (NHE3) located in the intestine, has already been registered in the United States as a promising agent to effectively prevent hyperphosphatemia in advanced CKD by means of reduced GI absorption [15,16,17]. This novel strategy may be of great importance since most therapeutic approaches used to date are of the limited efficacy concerning the “hard” endpoints, despite their positive impact on the CKD-MBD lab profile. Whether the new generation of phosphate-controlling agents would influence the patient outcome remains to be demonstrated. The latest data on pharmacological treatment of diabetes additionally challenged the traditional view on the role of phosphate homeostasis on outcome. Namely, it has been shown that SGLT2 (Sodium-glucose co-transporter type 2) inhibitors (such as dapagliflozin) promote phosphate retention, increase serum FGF23 and PTH, and decrease serum 1.25 (OH)_2_D_3_ [18,19]. Although this change in a biomarker profile clearly goes in the “wrong” direction when mineral metabolism is considered, SGLT2i are both cardio- and renoprotective and life-saving drugs for patients with and without diabetes and across CKD stages 1–4 [20,21,22,23,24,25,26,27,28].

## 3. Interactions between Glucose and Phosphate Homeostasis

Chronic inflammation, the hallmark of diabetes, obesity and CKD, contributes to the synthesis of FGF23 [29]. It has been demonstrated in animal models that insulin inhibits FGF23 synthesis. Bär et al. analyzed the relationship between plasma insulin concentration following oral glucose load and FGF23 in healthy volunteers, showing an inverse correlation between these two hormones. Such a relationship may suggest a potential impact of hyperinsulinemia on renal phosphate retention [30]. Garland et al. have demonstrated that insulin resistance is significantly associated with FGF23 increase in multivariable linear regression analysis—HOMA-IR (Homeostatic Model Assessment for Insulin Resistance) and eGFR (estimated glomerular filtration rate) decline were the only parameters out of broad spectrum of bone-turnover biomarkers, indices of inflammation and “classical” parameters reflecting the risk of atherosclerosis that influenced serum FGF23 in patients with CKD stages 3–5 [31]. Similar results were obtained by Hanks et al., who found an independent relationship between FGF23 and several indices of insulin resistance, which were much more apparent in subjects with normal kidney function as compared to patients with CKD stages 3–5; in the same study the authors confirmed a strong relationship between analyzed interleukins (IL6, IL10), C reactive protein and serum FGF23 [32]. Animal experiments have proven an essential role of phosphate in the normal secretion of insulin by pancreatic beta cells [33]. This finding has also been confirmed in humans—Haap et al. found that in 881 healthy subjects (non-obese, without diabetes) serum phosphate and glucose are inversely correlated and that serum phosphate was correlated positively with insulin sensitivity, but not with insulin secretion [34]. On the other hand, incidence of the type two diabetes was significantly higher in those subjects from the group of 71,270 participants followed in the French E3N-EPIC (Etude Epidémiologique auprès de femmes de l’Education Nationale study) cohort, who ingested diet with high phosphate content. In second, third and fourth quartile of phosphate intake there was a progressive and significant increase in the hazard ratio of T2D (1.18, 1.41., 1.54 vs. first quartile, respectively; with all increases being statistically significant). It is worth to mention that the mean intake of phosphorus in this study equaled 1477 ± 391 mg/day, which should be considered quite high (almost fifty percent higher than an average recommended daily intake). Neither GFR value nor serum creatinine were provided in the paper, but due to the population-based, observational design and very low prevalence of comorbidities (for example hypertension present in less than 13.5%), normal or near-normal renal function among study participants could be assumed [35]. This study highlights the importance of phosphate intake as another lifestyle-related risk factor for the development of T2D, probably not appreciated by most practitioners.

## 4. Impact of Phosphate on CV Disease and Outcome in General Population, CKD and Diabetes

Diabetic kidney disease (DKD) is generally accepted as the leading cause of CKD and ESRD [36]. CV disease (CVD) is a leading cause of mortality among CKD patients. The etiology of CKD matters when CV consequences are considered—a strong interaction exists between the underlying cause of CKD, indices of renal injury (such as: albuminuria, reduced GFR) and the risk of CVD. In other words, patients with the comparable degree of albuminuria and/or GFR, but suffering from diabetes, have a much higher risk of CV events and CV death as compared to those who developed CKD secondary to glomerulonephritis or polycystic kidney disease [37,38]. Profound endothelial dysfunction, resulting from hyperglycemia, hyperlipidemia, chronic inflammation, oxidative stress, products of bacterial metabolism resulting from gut dysbiosis and many other factors, triggers and then promotes the progression of both atherosclerosis (predominantly intimal pathology) and Mönckeberg sclerosis (involving media layer) in patients with diabetes and especially with DKD [39,40,41,42]. Better metabolic control of diabetes may significantly slow down the progression of micro- and macrovascular complications of diabetes, essential in the development of both CVD and CKD (DKD). Pathologic vascular calcification can already be found in young patients with ESRD treated with dialysis, in whom other “classical” risk factors of CVD, such as hypertension, atherogenic dyslipidemia or smoking, may be absent. Media calcification is induced by hyperphosphatemia, hypercalcemia, chronic inflammation and high parathyroid hormone concentration. We were among the first to describe the prevalence, advancement and progression of coronary artery calcification in patients treated with peritoneal dialysis [43,44]. Diabetic milieu strongly promotes these processes and, in diabetes, high glucose and high phosphate act synergistically [45]. Calcium and especially phosphate serve not just as substrates for deposition within vessel walls but also as potent inducers and regulators of vascular calcification (by some authors also described as ‘ossification’ due to similarities to the bone development process). PIT1(Sodium-dependent phosphate transporter 1) transporters allow for phosphate entry into vascular smooth muscle cells (VSMC) and trigger several intracellular events that finally result in trans-differentiation of VSMC into osteoblast-like cells [46,47]. VSMC with phenotype shift toward osteoblasts release extracellular vesicles (including exosomes and microvesicles), thus initiating the deposition of mineral content within the media layer. Extracellular vesicles are depleted with inhibitors of calcification (such as, for example fetuin A or matrix Gla protein) and may serve as the “nuclei of calcification” [47,48]. The infiltration of the media layer by macrophages and cytokine release from these cells further augment and promote calcification. Osteoblast-like cells and infiltrating macrophages lead to extracellular matrix (ECM) remodeling of vessel wall, further contributing to hydroxyapatite deposition [49]. The calcification of intima (atherosclerotic plaque) is considered a phenomenon secondary to atherosclerosis, although there is a strong interaction between these two processes—media calcification promotes and accelerates intimal calcification.

Hyperphosphatemia is linked to an increased risk of death across all ranges of CKD. For example in the meta-analysis published in JAMA in 2011 (4651 patients with non-dialysis dependent CKD) each 1 mg/dl of serum phosphate above the upper normal range was linked with 35% increment in the risk of death, although some studies failed to demonstrate an impact of serum phosphate on an outcome of patients with CKD [11,50,51,52,53]. Correlations can be found between serum phosphate (including high normal level) and risk of CV events also in patients with coronary artery disease without CKD [54,55,56]. Several mechanisms counteract processes of vascular and soft tissue calcification—among them, fetuin A remains the key inhibitor. Interestingly, this protein, considered unequivocally beneficial for protection against vascular calcification, leads to insulin resistance and may in fact aggravate the “proatherogenic” milieu in diabetics at the tissue level [57]. Indeed, Yilmaz et al. in the group of 82 patients with T2D demonstrated a strong relationship between high serum fetuin A and the advancement and progression of diabetic retinopathy [58]. This anti-calcifying protein has also been demonstrated to induce inflammation and lipotoxicity in podocytes, leading to podocyte death—podocyte damage and podocytopenia constitute the key pathologic background for development of DKD [59]. To further complicate the story of fetuin A in diabetes and metabolic syndrome, it has been suggested that this protein seems to protect from fibrosis and progressive liver damage in patients with non-alcoholic fatty liver disease [60].

The literature concerning the impact of high phosphate diet on patient-oriented or surrogate CV end-points is limited to the handful of publications [61,62,63,64]. In the most important study, Chang et al. found that an incremental intake of phosphorus above 1.4 g/day and phosphorus density above 0.35 mg/kcal were associated with an increasing risk of all-cause mortality (phosphorus density was defined as a ratio between dietary phosphorus amount to the total daily calorie intake). High phosphorus density (>0.35 mg/kcal) but not phosphorus intake was also associated with increased CV mortality. The J-curve phenomenon was noticed in this NHANES III (National Health and Nutrition Examination Survey III) population-based population study—mortality was increased also in subjects with the lowest phosphorus consumption. Patients with diabetes and CKD were excluded from this important trial, and kidney outcomes were not addressed [61]. Another NHANES III-based analysis that included only patients with CKD (baseline mean eGFR of 49.3 ± 9.5 mL/min/1.73m^2^; depending on subgroups analyzed; 14% to 28% suffered from diabetes) did not confirm the above findings—Murtaugh et al. failed to demonstrate an association between phosphate intake and outcome and found no association between phosphate intake and serum phosphorus. As in the study of Chang et al., the authors did not look at the CKD progression [62].

## 5. Impact of Phosphorus on Development and Progression of Non-Diabetic and Diabetic CKD

Interestingly, in early diabetes (i.e., with well-preserved kidney function) patients may experience phosphate depletion rather that phosphate accumulation since in the setting of hyperglycemia proximal tubular cells are less efficient in phosphate reabsorption and hyperphosphaturia may develop [65,66]. Phosphate depletion may result in impaired cellular energy balance which resemble tissue ischemia. It is especially the case in patients with poorly controlled diabetes and hyperglycemia and affects primarily cells in which the glucose uptake is not controlled by insulin. Such a situation may activate pathways stimulating the synthesis and release of erythropoietin, an essential hormone contributing to the defense against ischemia. An increase of erythropoietin synthesis despite well preserved serum hemoglobin has been described in patients with diabetes [66,67].

In some studies, hyperphosphatemia (or even high-normal serum phosphate) has been identified as an independent predictor of CKD progression in patients with CKD of different etiology as well as the incidence of de novo CKD in the general or non-CKD population (for example NHANES III or Framingham Heart Study cohorts), although some observations do not confirm this notion [50,68,69,70,71]. KEEP (the Kidney Early Evaluation Program) Investigators did not find an association between serum phosphate and all-cause mortality or the risk of end-stage renal disease in their cohort of 10,672 patients observed for median period of 2.3 years. Forty seven percent of patients included in this trial suffered from diabetes—when the results were stratified according to diabetes status, serum phosphate remained non-significant as a predictor of mentioned outcomes [50]. Xiang et al. made an interesting observation of two patient groups with biopsy-proven CKD (stages 1 to 4): 591 with biopsy-proven DKD and 957 with IgA nephropathy (i.e., most common secondary and primary glomerular disease in the general population); renal function was well-preserved at the start of the follow-up and equaled 73.3 ± 30.6 mL/min/1.73 m^2^. The renal end-points in this study were defined as permanent decrease of eGFR to less than 15 mL/min/1.73 m^2^ (i.e., CKD stage 5) and the need for dialysis commencement or renal transplantation. Both groups were divided into quartiles depending on the baseline phosphate level. Of note, patients in the highest quartile of serum phosphate in both groups most frequently reached the renal end-points (IgA patients—26.8% and DKD patients—57.7%) over the median follow-up period of 40 months. The effect of serum phosphate on CKD progression rate was independent from baseline eGFR. Patients with biopsy-confirmed DKD in the 4th quartile of baseline phosphate were characterized by almost five times higher risk to achieve renal end-point as compared to those from the first quartile (fully adjusted HR (Hazard ratio) in Q4 vs. Q1 (quartile 4 vs quartile 1) equaled 2.88; *p* = 0.024 for DKD and 1.68 (non-significant) for IgA nephropathy)) [72]. The rate of CKD progression differs depending on an underlying diagnosis of renal disease; however, the progression of CKD is fastest in DKD (as compared for example to cystic kidney disease, primary glomerular disease or other nephropathies) [38]. Patients with DKD are characterized with higher serum phosphate and decreased phosphaturia as compared to those with other underlying causes of CKD but corresponding values of GFR. Other biomarkers characterizing CKD-MBD also become abnormal earlier in the course of DKD as compared to other nephropathies—this is the case of serum FGF23 or 25 (OH) D3 [73].

FGF23, one of key indices of disturbed calcium—phosphate homeostasis was repeatedly demonstrated to predict all-cause and CV mortality in different populations, including also the general population, although its independent predictive value has been questioned recently [74,75,76]. Several studies have found that high FGF23 can also predict the progression of CKD in non-diabetic patients with various underlying causes of nephropathy, including glomerulonephritis, interstitial nephritis, hypertensive renal disease, autosomal dominant polycystic kidney disease and non-diabetic CKD of unknown etiology [77,78,79]. Concerning DKD, it is worth mentioning that the results of the study performed by Isakova et al., who analyzed the impact of baseline FGF23 measured early in the course of a disease, that is, when eGFR equaled 90.9 ± 22.7 mL/min/1.73 m^2^ on the incidence of CKD (defined as permanent reduction of eGFR to less than 60 mL/min/1.73 m^2^) in a subset of patients included in the ACCORD trial (Action to Control CV Risk in Diabetes). Applying different prediction models, the authors found that baseline FGF23 does not predict the incidence of CKD in patients with diabetes, although they suggested the contribution of this parameter to CKD progression, once already established [80]. These results seem to remain in agreement with earlier findings published in the same journal (Clinicaj Journal of the American Association of Nephrology) by Titan et al.—these authors found that FGF23 predicts the combined outcome defined as death, doubling of serum creatinine or a need for dialysis in the cohort of diabetic patients with microalbuminuria and reduced baseline eGFR (i.e., with established DKD) [81]. Agarwal et al. used a broad spectrum of plasma and urine biomarkers and tested their ability to predict the progression of DKD. Out of 17 urine and 7 plasma biomarkers analyzed (reflecting key processes involved in the progression of CKD: inflammation, fibrosis, angiogenesis, tubular function and injury, CKD-MBD) FGF23 was documented as the strongest predictor of end-stage renal disease in a cohort of diabetic patients with established DKD (eGFR of 43 ± 13 mL/min/1.73 m^2^ and mean albuminuria equalining 405 ± 673 mg/g) [82].

In our opinion, the equivocal results obtained in studies addressing the impact of serum phosphate and FGF23 on renal and cardio-vascular outcomes results from the construction of the models and the choice of variables taken for analysis. For example, in many observational trials that focused on phosphorus and calcium-phosphate homeostasis, biomarkers proteinuria or albuminuria were not appreciated enough. In a large cohort of patients with various proteinuric kidney diseases (including group of patients with DKD), it has been documented that proteinuria (albuminuria) leads to phosphate retention independent on the GFR value and despite the increase in plasma FGF23 (although the association between albuminuria and phosphate retention was stronger for diabetics and patients with GFR of less than 45 mL/min/1.73 m^2^). In the experimental part of the cited study, de Seigneux et al. showed that rats with puromycine aminonucleoside-induced nephrotic proteinuria are characterized with significantly increased NaPi-IIa tubular expression, decreased renal klotho protein expression and decreased phosphorylation of FGF receptor substrate 2α, responsible for the tubular phosphaturic effect of FGF23 [83]. These findings highlight the critical role of proteinuria in the development of “renal resistance” to FGF23. Since albuminuria and proteinuria are the well-identified and pivotal risk factors for the development of CV events and CKD progression, it seems likely that, at least to some extent, the impact of phosphate and FGF23 on prognosis may depend on its relationship with albuminuria.

Despite existing controversies, the association between serum phosphate and the risk for the development and progression of diabetic and non-diabetic kidney disease seems to be well documented. However, the data indicating the link between dietary phosphorus intake and renal outcomes (usually defined as: doubling of baseline serum creatinine, reduction of baseline GFR by 50%, fall of GFR below 60 and below 15 mL/min/1.73 m^2^ or need for the renal replacement therapy) are very scarce. Yoon et al. analyzed phosphorus intake based on dietary survey in patients involved in the Korean Genome and Epidemiology Study. The study has an observational (non-interventional) design and involved in total 6719 subjects, out of whom 873 were diabetic. Phosphorus intake was analyzed in relation to calorie consumption (phosphorus density) and was expressed in mg/kcal. De novo (incident) CKD developed in 32.4% diabetic and in 13.5% non-diabetic participants over 100 ± 41.1 and 115.4 ± 33.8 months of follow-up, respectively. The authors found that the highest quartile of phosphorus density was associated with significantly increased risk of incident CKD in diabetic patients (HR 1.68, 95% CI 1.08–2.63, *p* = 0.02 vs. Quartile 1) and such a relationship was not found in non-diabetic participants. The same was also true for survival—a significant impact of the highest phosphorus intake density was noticed only for diabetics. It is worth mentioning that the cut-off of phosphate intake density from which the risk started to increase equaled 0.45 mg/kcal in diabetic patients [84].

## 6. Therapeutic Interventions to Lower Serum Phosphorus for Cardio- and Nephroprotection

Most of the data concerning the therapeutic role of dietary phosphorus restrictions and the efficacy and usefulness of phosphate binders come from the “renal” literature. Abnormalities in mineral metabolism occur early in the course of CKD, but overt hyperphosphatemia is apparent when GFR falls to the range of 25–40 mL/min/1.73m^2^ [85]. In CKD patients it should be attempted to keep serum phosphate concentration within the normal range and in those patients with CKD stage 3a–5 not on dialysis, in whom phosphate level exceeds the upper normal value, the treatment should be started. The time to start phosphate-lowering treatment has not been precisely defined, however, decisions should be made based on the serial measurements of this parameter and persistently high or increasing serum phosphate should prompt treatment. Context of other parameters, such as calcium, PTH, 25 (OH) D3 and 1.25 (OH)2D3, should also be taken into consideration in decision-making (FGF23 is not measured routinely in clinical practice). Certainly, there is no need to lower serum phosphate below the lower normal value [86]. Until now, none of the tested strategies to reduce serum phosphate were demonstrated to change outcome in patients with CKD, both those in the pre-dialysis period and those on maintenance dialysis—this statement holds true for both dietary interventions and phosphate binders. In fact, none of the methods of intervention applied to treat CKD-MBD proved theirits efficacy in terms of changing patient outcome (including “native” vitamin D, “active” vitamin D compounds and calcimimetics). In principle, two methods of phosphate control are applied in patients with non-dialysis dependent CKD—decreased dietary phosphorus intake and oral phosphate binders (vitamin D compounds may actually increase serum phosphate since they increase phosphate absorption from GI). It is recommended in CKD patients to moderate their phosphate intake in order to limit the risk of hyperphosphatemia. Since phosphorus is abundantly present in many nutritional products, phosphate—restricted diet may increase the risk of other nutrient deficiencies. On the other hand, in the more advanced stages of CKD, a diet with reduced protein intake is also recommended, which may help to achieve the lowered phosphate intake [87]. Dietary interventions, as well as phosphate binders, were shown to correct several biochemical parameters of CKD-MBD, including serum phosphate and parathyroid hormone, however until now such an impact has not been shown when clinically meaningful end-points were considered [88].

Serum phosphate within the normal range is also recommended for patients on dialysis. As in non-dialysis CKD patients, the reduction of daily phosphate intake to less than 900 mg is recommended. From the practical point of view, dietary counselling may be needed in order to avoid the worsening of nutritional status (the risk of malnutrition is an important issue in this patient group since CKD—especially treated with dialysis—is a catabolic disease). Contrary to the advanced stages of non-dialysis CKD, when protein intake reduction may be considered beneficial as being nephroprotective, dialysis patients need a rather high protein diet to compensate for ongoing catabolism and amino acid losses into dialysate during hemodialysis (and amino acids and protein—during peritoneal dialysis). As in pre-dialysis patients, phosphate binders become the next step to control serum phosphate. Observational studies demonstrated that the use of phosphate binders may lower mortality of patients treated with dialysis, and that this is especially true for calcium-free phosphate binders [89,90,91,92]. As for today, the long term safety and impact on patient-oriented clinical outcomes of new iron-based phosphate binders cannot be established yet, however some safety signals on their adverse effects can be noticed from experimental trials [93,94]. Carefully performed and comprehensive analyses have demonstrated that one phosphate binder may (at best) be better than another in terms of particular side effects or the onset of surrogate end-points. However, no clinically meaningful benefits of any phosphate binder have been documented in randomized controlled trials on such outcomes as CV death, myocardial infarction, stroke or fractures not only different phosphate binders when comparing, but also in studies comparing phosphate binders vs placebo [95,96]. Clinically unproven and rather theoretical benefits of phosphate binder use are additionally offset by side effects of these drugs—historically, aluminum toxicity was the leading one, and—when toxicity of these compounds was appreciated—excess of calcium intake after switch into calcium-containing phosphate binders became the leading safety issue. Certain toxicities of lanthanum carbonate have also been described. The use of calcium—containing phosphate binders (represented by calcium acetate and calcium carbonate)—although quite effective in terms of phosphate binding and in normalization of serum parathyroid hormone, resulted in increased absorption of calcium from the GI (especially when used with vitamin D compounds) and fueled vascular, valvular and soft tissue calcification. In addition, excess calcium leads to the oversuppression of parathyroid hormone synthesis and release, which is associated with so called “low turnover” bone disease and increased risk of bone symptoms, including fractures [97]. Sevelamer carbonate and sevelamer hydrochloride appear the safest drugs in the field, but they also remain “neutral” in terms of hard clinical end-points (despite being more effective in slowing down the progression of vascular calcification as compared to calcium-containing phosphate binders) [96].

In patients treated with renal replacement therapies, phosphate elimination with dialysis also contributes to phosphate balance (although at the usually prescribed “dose” of dialysis, i.e., 12–15 h per week, divided into three separate sessions phosphate clearance is insufficient to control hyperphosphatemia). In a few existing programs of daily (or nightly), extended hemodialysis phosphate clearance by dialysis improves significantly and allows us to abandon not only phosphate binders but also dietary restrictions in phosphate intake [97,98].

Concerning nephroprotective measures of both mentioned strategies (i.e., restrictions in dietary phosphorus and phosphate binders), no good quality data can be found in the literature. The reference document for nephrologists in this field, the KDIGO 2017 Clinical Practice Guideline Update for the Diagnosis, Evaluation, Prevention, and Treatment of Chronic Kidney Disease-Mineral and Bone Disorder (CKD-MBD) does not address at all the issue of possible nephroprotection using any phosphate-oriented intervention [87]. It is worth to mention, that the document published recently in order to address enormous progress in the field of diabetes and DKD stemming from the results of landmark GLP1 receptor agonist and SGLT2 inhibitor trials, namely the KDIGO 2020 Clinical Practice Guideline for Diabetes Management in Chronic Kidney Disease does not mention the role of a low-phosphate diet as nephroprotective strategy. Dietary guidelines, including avoidance of processed food in order to limit sodium intake or moderate reduction in protein intake may actually help to achieve the goal of reduced intake of phosphorus, but such a goal is not expressed directly throughout the whole document. Interestingly, the same document does not mention vitamin D, the notoriously overused OTC (Over the counter) drug, as having any value for nephroprotection in patients with diabetes [99]. Phosphate toxicity and the need to limit phosphate ingestion is not mentioned in any section of the latest ADA 2021 recommendations for the diagnosis and treatment of diabetes, including the lifestyle and diet, prevention and treatment of CVD or microvascular complications. The same holds true for vitamin D [100].

## 7. Conclusions

Phosphate, derived mostly from preserved food and animal-based products, is identified as one of the important nutrients contributing to CV and renal disease, and its toxicity is of particular importance in patients with CKD, the clinical entity characterized with profound derangement of mineral homoeostasis. A closer look at the available literature shows that the data on the influence of dietary phosphorus on CV and renal outcomes are very limited and this statement is also true for diabetic kidney disease. Although (despite some controversies) it seems reasonable to conclude that high serum phosphate and other biomarkers of mineral homoeostasis predict the onset and progression of DKD, the data confirming a direct link between these end-points and high dietary phosphate intake are scarce. Serum phosphate level depends on several variables and phosphate intake is only one of them. The story of dietary phosphorus and its contribution to outcome in our opinion resembles the story about vitamin D used as a “panacea” for all existing diseases. High plasma vitamin D (25(OH)D3) is universally identified as a predictor of excellent outcome, but—as for today—no trial has documented a beneficial impact of vitamin D supplementation on any patient-oriented end-point (including a lack of clear benefit on the risk of osteoporosis in post-menopausal women). Serum 25[OH]D3 or 1.25(OH)2D3, tissue effects of “active” vitamin D and the onset of “hard-endpoints” depend on a countless number of variables apart from vitamin D intake. The situation with dietary phosphorus (universally considered as toxic—and with confirmed toxicity in basic research) and clinical end-points in a real life is similarly complicated. In our opinion, it seems reasonable to limit phosphorus intake for several reasons (phosphorus-rich products are also rich in sodium, protein, additives). Since phosphorus content is high in processed food, high phosphorus intake may also reflect the lifestyle, education level, cultural issues, and so forth—these variables are not universally taken into consideration in the studies [49]. On the other hand, healthy food products such as nuts, beans and fish are also rich in phosphorus. As indicated by Yoon et al. and the NHANES III investigators, the caloric context of phosphorus intake may be of great importance—neglecting the amount of calories ingested with a particular amount of phosphorus may be the source of misinterpretations of its impact on kidney health [49,72]. In conclusion, as for today, no clear recommendations based on the firm clinical data can be provided in terms of phosphorus intake aiming to prevent the incidence or progression of diabetic kidney disease.

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
