# Peer review of "Dietary Phosphorus as a Marker of Mineral Metabolism and Progression of Diabetic Kidney Disease"

_nutrients, 2021, doi:10.3390/nu13030789_

Round 1

Reviewer 1 Report

The authors have presented a detailed and well organized review of role of phosphorus in diabetic kidney disease. I have a few minor comments

  1. The authors did talk about dietary sources of phosphate in the Conclusion section. It would also be helpful to include a statement in line 33 when talking about dietary phosphate.
  2. Line 29: Did the authors mean "bound" or "found" in this line?
  3. Line 43: extra space after elucidated
  4. Line 83: potentially modifiable variable, its impact...
  5. Line 140-141: please include p for linear trend if available in this reference
  6. Line 317: other renal outcomes?
  7. Line 436: scarce - spelling

Author Response

Thank you so much for all valuable comments. We did our best to address them in detail:

  1. Regarding comment No 1 and 6: we added respective fragments to follow your suggestions and improve the quality of the text.
  2. Regarding comment no 2: we meant phosphate deposition in the bone. We changed this in the text to make it clear.
  3. We corrected all writing, typing and spelling errors you mentioned in your comments No 3, 4, 7
  4. Regarding comment No 5: p values are not provided for these hazard rations, all were statistically significant which can be concluded from the 95% Confidence Interval values. We added the statement concerning significance and decided not to add the 95% CI values – several additional numbers might make the text difficult to follow.

Reviewer 2 Report

Dear editors:  

 It is a great honor and pleasure for me to be invited as the reviewer for this review article. Winiarska et al. comprehensively reviewed the role of dietary phosphorus in mineral metabolism and progression of diabetic kidney disease. This study topic is interesting and important, attributing to Prof. Stompór’s long-term efforts and contributions in this scientific field. The review work is comprehensive. I have a number of comments concerning this study:

  1. Title: Dietary phosphorus is a singular noun. The title might be rephrased as “Dietary phosphorus as a marker of mineral metabolism and progression of diabetic kidney disease”
  2. Line 41: The term “NaPi-Iib” should be expressed as “NaPi-IIb”.
  3. Line 43 should be rephrased as “… fully elucidated. It seems that these pathways are calcitriol independent…”
  4. Line 67: Since the term “cardiovascular” was mentioned in the text for the first time, the abbreviation “cardiovascular (CV)” should be used here. Thus the term “cardiovascular” should be replaced by “CV” in Line 69, 86, 94, 96, 149, 152….
  5. Line 83: It would be better that the term “factor” substitutes for “variable”.
  6. Line 84: From the perspective of nephrologist, the key therapeutic strategy in CKD is not merely reduction of dietary phosphate intake but also use of phosphate binders [Reference 12]. Proper amount of phosphate intake is important to avoid malnutrition and protein-energy wasting. In contrast, hypophosphatemia due to inadequate phosphate intake and malnutrition increases the mortality rate in dialysis patients [Combined alkaline phosphatase and phosphorus levels as a predictor of mortality in maintenance hemodialysis patients. Medicine (Baltimore). 2014 Oct;93(18):e106.]. Thus hypophosphatemia and inadequate phosphate intake could counteract the protective role of reducing dietary phosphate intake. The sentence could be rephrased as “reduction of excess phosphate intake with phosphate binders has long been recognized as one of the key therapeutic strategies in CKD.” Only excess phosphate intake without adequate removal can lead to hyperphosphatemia and associated clinical events. The calcium-based phosphate binders are common and useful methods of phosphate removal. Thus daily calcium amount in foods, fortification and supplementation is a potential confounder. However, it could not be precisely evaluated and adjusted in most observational studies.
  7. Line 152: Since the term “CVD” was used here, “cardiovascular disease” in Line 165 should be replaced by “CVD”.
  8. Line 161: In fact, atherosclerosis is a specific kind of arteriosclerosis, but these terms are often used interchangeably. Sentence in Line 162 should be rephrased as “atherosclerosis (predominantly intimal pathology) and Mönckeberg's sclerosis (involving media layer) in patients with diabetes and…”
  9. Line 168: “are” absent => “may be” absent
  10. Line 189: Hyperphosphatemia is linked to the increased risk of death across all ranges of CKD. However, the reference for patients with dialysis is not cited in this section. The population in JAMA study (Reference 40) was non-dialysis dependent CKD patients. I suggested some articles about the association between serum phosphate levels and mortality could be cited here, e.g., Combined alkaline phosphatase and phosphorus levels as a predictor of mortality in maintenance hemodialysis patients. Medicine (Baltimore). 2014 Oct;93(18):e106. It is a pity that the authors did not address the impact of hypophosphatemia and inadequate phosphate intake on CV outcomes here.
  11. Line 195: The sentence seems redundant. Cardiovascular events includes mortality and non-fatal outcomes. The words “cardiovascular events and cardiovascular mortality” could be more concisely with “ CV events”.
  12. Line 458-464: This paragraph should not be presented here as an end of the conclusion. The paragraph may be inserted in Line 425 or further discussed in a new section.
  13. Please add the abbreviation list after the body text, and all the abbreviations in the text should be presented here.

Thank you for giving me the opportunity to review this interesting article.

Author Response

We deeply appreciate your input into our work by making all these valuable comments. As a corresponding author I would like to acknowledge the very kind comment of the Reviewer addressing my personal efforts and contribution to the research.

We made the following changes in a text:

  1. We changed the title of the paper according to your suggestion (comment 1).
  2. We corrected all writing, typing and spelling errors you mentioned in your comments No 2, 3, 4, 5, 7, 8, 9. We also changed indicated terms into abbreviations throughout the whole text.
  3. Your comment No 6: thank you very much indeed. After the reading of an original version the reader might have impression that hyperphosphatemia is the only harmful event related to phosphate homeostasis disorders. We added a respective statement concerning the risk associated with low serum phosphate, mostly (but not only) due to malnutrition. To document this statement we cited a milestone study of Chang JF, et al, Medicine (Baltimore). 2014.
  4. Concerning comment No 10: you are perfectly right. We were more focused in our review on earlier stages of CKD, however when mentioning all stages of CKD the statement and citation on ESRD and dialysis should be added. For this purpose we decided to cite the paper of Chang JF, et al, Medicine (Baltimore). 2014.
  5. Concerning comment No 11: we rephrased this sentence accordingly.
  6. Concerning comment No 12: indeed, these sentences should not finalize the paper. We moved them into the paragraph 2, when we discuss new challenges in our understanding of phosphate metabolism disorders and their treatment.
  7. We attempted to add the complete list of abbreviations after the body of text. It was unclear to us whether your suggestion was to remove all abbreviated terms from the text or leave them when used for the first time. In the R1 version we left them only when used for the first time.

Reviewer 3 Report

The authors conduct a literature review and aimed to discuss the data concerning chronic kidney disease (CKD)-Mineral and bone disorders(MBD) in patients with diabetes, and to analyze the possible link between hyperphosphatemia, certain biomarkers of CKD-MBD and high dietary phosphate intake on prognosis in patients with diabetes and diabetic kidney disease (DKD).

Comments:

It would be convenient for the authors to reflect in the Abstract section what are the conclusions derived from the review to allow a rapid assessment.

A comparison infographic or Box/Table that summarizes the information can make it a lot easier for readers.

Author Response

We appreciate your valuable comments.

  1. We added the phrase helpful for a rapid assessment of the paper’s content into the Abstract.
  2. We attempted to summarize the key issues of our review in the form of infographic (graphical abstract). We hope it will be helpful for potential readers.

Round 2

Reviewer 3 Report

No further comment